# G-band Radar for Water vapor and Arctic Clouds (GRaWAC): novel insights on Arctic water vapor, clouds and precipitation

Sabrina Schnitt<sup>1</sup>, Mario Mech<sup>1</sup>, Jens Goliasch<sup>2</sup>, Thomas Rose<sup>2</sup>, and Susanne Crewell<sup>1</sup>

<sup>1</sup>Institute for Geophysics and Meteorology, University of Cologne, Cologne, Germany

**Correspondence:** Sabrina Schnitt (s.schnitt@uni-koeln.de)

#### Abstract.

Clouds are a central component in the complex interplay of feedback processes driving amplified warming in the Arctic. While state-of-the-art radar observations operating at X-, Ka-, or W-band provide detailed measurements of both hydrometeor distribution and cloud dynamics based on radar moments and spectra, current microphysical retrieval approaches are limited at small particle sizes ubiquitously present in Arctic clouds. Expanding observations to the G-band (110-300 GHz) can bridge this limitation and provide additional information on the vertical in-cloud water vapor distribution through the Differential Absorption Radar (DAR) method. Here, we introduce the dual-frequency, Dopplerized frequency-modulated continuous-wave (FMCW) G-band Radar for Water vapor and Arctic Clouds (GRaWAC), which operates simultaneously at 167.3 GHz and 174.7 GHz. GRaWAC uniquely combines DAR and Doppler capabilities in a bi-static system, achieving a sensitivity of -43 dBZ at 1 km range and a 1 s integration time with a vertical resolution of 20 m. GRaWAC's flexible design enables operational usage from ground, ship, or aircraft. By applying the Differential Absorption Radar (DAR) approach, we retrieve in-cloud and in-precipitation water vapor profiles, as well as partial column water vapor in all-sky conditions, when deployed from aircraft. In order to highlight GRaWAC's potential, we present first measurements from Arctic deployments at (1) the German-French AWIPEV research base, Ny-Ålesund, Svalbard, with water vapor profiles derived at 60 s temporal and 200 m vertical resolution, and (2) aboard the AWI Polar 6 aircraft with profiles' vertical and horizontal resolution of 200 m and 1.6 km, respectively.

We find that ground-based retrievals capture lower-tropospheric moistening well. We find a RMSD of 0.5 gm<sup>-3</sup> compared to coincident radiosonde profiles, which increases to up to 2 gm<sup>-3</sup> in the case of differential scattering and liquid attenuation systematically affecting the retrieval approach. The temporal evolution of a mixed-phase cloud deck is well represented in Doppler spectra at 167.3 GHz, as characteristic bi-modal peaks form when supercooled liquid and ice coexist. During airborne operations, a case study shows that GRaWAC and a dropsonde water vapor profile agree remarkably well (within 0.5 gm<sup>-3</sup>). Retrieved IWV is sensitive to instrument calibration, and shows an RMSD of 1.1 kgm<sup>-2</sup> compared to the statistics of dropsondes across different surface types, including sea-ice. While further investigation and retrieval development is needed to mitigate differential scattering and liquid attenuation effects on retrieved water vapor profiles, e.g., through a synergistic optimal estimation retrieval including microwave radiometer and W-band radar, our results highlight GRaWAC's potential to bridge the observational gap between Arctic cloud microphysics and thermodynamics.

<sup>&</sup>lt;sup>2</sup>RPG Radiometer Physics GmbH, Meckenheim, Germany

#### 1 Introduction

Radars serve as the backbone for observing Earth's atmospheric water cycle. Centimeter-wavelength radars have been in use since the 1950s to monitor precipitation in weather service radar networks. Thanks to increased sensitivity to cloud droplets and ice hydrometeors, mm-wavelength cloud radars have emerged in the past decades as state-of-the-art instrument to study cloud microphysics and precipitation processes (Lhermitte, 1990; Kollias et al., 2007) continuously at ground-based supersites (e.g., Gierens et al., 2020), during designated research studies on aircraft (e.g., Mech et al., 2019), or globally from space with CloudSat (Stephens et al., 2018) or the Cloud Profiling Radar (CPR, Kollias et al., 2014) on EarthCARE (Wehr et al., 2023).

In the Arctic, clouds, water vapor, and precipitation are at the heart of feedback processes contributing to Arctic Amplification (Tan and Storelymo, 2019; Wendisch et al., 2024). Arctic low-level clouds are ubiquitously mixed-phase (Mioche et al., 2015; Gierens et al., 2020), resilient (Morrison et al., 2012), and, depending on their phase distribution, strongly affect the radiative balance (e.g., Curry et al., 1996). Cloud radar observations are, therefore, crucial for both improving the quantification and enhancing the understanding of key processes driving mixed-phase cloud evolution and precipitation formation. While observed moments are used to derive in-cloud motion and macro-physical parameters (e.g., Shupe et al., 2008), cloud radar spectra exhibit characteristic bi-modal shapes which can be used to infer the vertical phase distribution (Shupe et al., 2004; Verlinde et al., 2013; Kalesse et al., 2016). Still, cloud radar measurements, even when combined with auxiliary lidar and radiation measurements, do not provide the full picture of cloud characteristics. While multi-frequency measurements help to constrain particle size, today only a limited set of radar frequencies is used, which leads to limitations at smaller particle sizes (e.g., Battaglia et al., 2014). Furthermore, to understand the formation and decay of clouds, simultaneous observations of cloud and in-cloud water vapor are necessary.

Expanding cloud radars to the G-band (110-300 GHz) offers exciting possibilities for improving the understanding of cloud and precipitation formation and evolution (Battaglia et al., 2014), including Arctic conditions. Only recently, hardware development has advanced sufficiently to build instruments with satisfactory stability and sensitivity, such as the Vapor-in-Cloud Radar (VIPR; Cooper et al., 2021) operating at 155, 167, and 174 GHz, the G-band Radar for Cloud Evaluation (GRaCE; Courtier et al., 2022) at 200 GHz, and, most recently, CloudCube (Socuellamos et al., 2024) at 239 GHz.

Higher frequencies are more sensitive to smaller droplets and ice hydrometeors (Lhermitte, 1990) when observing the same volume with the same transmitted power (Mead et al., 1989; Battaglia et al., 2014). In the G-band, non-Rayleigh scattering effects occur at smaller hydrometeor sizes where Rayleigh scattering is still present in Ka- or W-band (Lamer et al., 2021). This results in characteristic Mie notches shaping the Doppler spectrum (Courtier et al., 2022), which in turn can be used to retrieve vertical air motions more accurately than in state-of-the-art radars (Courtier et al., 2024). After correcting for air motion, Yurk et al. (2025) successfully retrieved Droplet Size Distributions (DSDs) and precipitation parameters in light warm rain conditions. McCusker et al. (2025) demonstrate improvements in ice microphysical parameters, such as Ice Water Content (IWC) and snowfall rates, compared to existing approaches.

When embedding G-band in a multi-frequency radar suite, the differential radar signals are several dBs higher than those of state-of-the-art combinations (Lamer et al., 2021; Socuellamos et al., 2024). This increased dynamic signal range of a Ka-G combination has been predicted to enhance the sensitivity of existing retrievals, such as those with a Ka-W combination, in areas of smaller hydrometeors (Battaglia et al., 2014). Based on field measurements with a Ka-G combination, improvements have been made for liquid water profiling in shallow clouds (Socuellamos et al., 2024) and for light warm rain microphysical parameters (Courtier et al., 2024).

Measuring at two frequencies along the wing of the 183 GHz water vapor absorption line allows for profiling water vapor in cloudy and precipitating columns using the Differential Absorption Radar (DAR) technique (Lebsock et al., 2015; Roy et al., 2018). Using the VIPR instrument with channels at 167 and 174.8 GHz, Roy et al. (2018) evaluate VIPR-derived profiles with more than 20 coincident radiosonde profiles and find a mean root-mean-square-difference (RMSD) of  $0.8 \, \mathrm{gm^{-3}}$ . Frequency-dependent hydrometeor scattering effects lead to systematic biases of  $2 \, \mathrm{gm^{-3}}$  or less. Battaglia and Kollias (2019) propose to use DAR for profiling relative humidity in ice clouds, mitigating differential scattering impacts in the retrieval by adding a third frequency to the retrieval. Embedding a third frequency in the retrieval accounts for differential hydrometeor scattering effects in the retrieved water vapor profile (Roy et al., 2020; Millán et al., 2024). Combining DAR with passive microwave radiometry has been shown to further advance full-column water vapor profiling methods (Schnitt et al., 2020). Operating VIPR from an aircraft additionally allows the retrieval of column water vapor amounts between the aircraft and the ground (Roy et al., 2022; Millán et al., 2024). IWV can be retrieved from ground Normalized Radar Cross Section (NRCS)  $\sigma_0$ . While Roy et al. (2022) find an RMSD of  $0.8 \, \mathrm{kg} \, \mathrm{m}^{-2}$  compared to dropsonde IWV in clear-sky conditions over the ocean, RMSD increases up to  $5 \, \mathrm{kg} \, \mathrm{m}^{-2}$  in cloudy conditions due to differential hydrometeor scattering and attenuation effects.

We present here the G-band Radar for Water vapor and Arctic Clouds (GRaWAC). GRaWAC is a Dopplerized frequency-modulated continuous wave (FMCW) radar transmitting and receiving simultaneously at 167.3 and 174.7 GHz. Its bi-static design mitigates problems with transmit/receive separation. For the first time, it combines DAR and Doppler capabilities in one instrument. Thanks to its versatile design, specifically designed for operation in harsh polar environments, GRaWAC enables continuous, automatic cloud and precipitation measurements from the ground, ship, or aircraft. The system was designed to achieve a sensitivity of better than -43 dBZ at a 1 km range and a 1 s integration time, with a vertical resolution of approximately 20 m, specifically to investigate Arctic clouds.

After giving a technical overview of the instrument (Sec. 2) and revisiting the DAR water vapor retrieval (Sec. 3), we illustrate first measurements in the Arctic performed at the German-French AWIPEV research base in Ny-Ålesund, jointly operated by the Alfred Wegener Institute Helmholtz Centre for Polar and Marine Research (AWI) and the French Polar Institute Paul Emile Victor (IPEV) (Sec. 4) and aboard AWI's Polar 6 aircraft (Sec. 5). We conclude the manuscript by summarizing current and future planned capabilities.

105

110

#### 2 GRaWAC Technical Overview

#### 2.1 Instrument Design

A Frequency-Modulated Continuous Wave (FMCW) radar employs frequency modulation of a so-called chirp to derive ranging information from the frequency difference between the transmitted and received signals. The DAR principle is realized in GRaWAC by continuously generating frequency chirps around two operating radio frequencies (RF), 167.3 and 174.7 GHz. Thanks to GRaWAC's bi-static antenna system (one for transmission and one for reception), both frequency signals are transmitted and received simultaneously using the same antenna. Once the chirps are reflected back to the instrument, they are received by an identical antenna placed next to the transmitting antenna (Fig. 1a) In this bi-static design, the separate transmit and receive antennas have a diameter of 500 mm each. The corrugated feed horns, in conjunction with the Cassegrain antenna, offer a half-power beamwidth of 0.36° and a gain of 54.6 dB.

Using this design approach, high isolation of the transmit and receive paths is achieved, albeit with the trade-off of non-ideal beam overlap. 90 % overlap is reached at 500 m distance from the radar. Generally, the radar measures from a distance of 100 meters. However, for distances less than 500 m away from the instrument, an overlap correction is applied to account for the height-dependent overlap of the transmit and receive beam volumes. Herein, the overlap loss is calculated as a function of range from the antenna separation distance, assuming a Gaussian-shaped beam profile.

In the receiver, the two frequency signals are separated by a frequency diplexer and down-converted by two independent mixers to an intermediate frequency ranging from 0.3 to 4 MHz. The mixer's Local Oscillator (LO) signals are generated by the transmitter stage (see Sec. 2.1) to form a homodyne system typical for FMCW radars. The same LO is also used for generating the transmit signal (see below). The down-converted and filtered signals are digitized using an analog-to-digital converter (ADC) board and post-processed by the Radar-PC.

Figure 1. (a) GRaWAC during test operations at the manufacturer and (b) technical block diagram.

125

## 2.2 Chirp generation and calibration

To generate the chirps, both frequency channels are referenced to a single local oscillator, with chirp generation achieved through direct digital synthesis. The chirps are fully synchronized and also serve as LO for receiver down-conversion mixers (see above). Two parallel up-conversion chains amplify and multiply the frequency chirps from 6.97 to 167.3 GHz and 7.27 to 174.7 GHz, respectively. The chirp generator produces a fundamental signal  $f_1$  around 6.97 GHz, which directly feeds the 167.3 GHz multiplication chain. From  $f_1$ , another chirp signal  $f_2$  (around 7.27 GHz) is derived via a phase lock loop (PLL) to drive the 174.7 GHz multiplier chain. This way, both frequencies are transmitted and received simultaneously and in parallel without the need for switching between them.

The exact bandwidth and frequency ramp of the signals depend on the configuration of the chirp sequence in terms of range resolution and Doppler Nyquist range. Typical unambiguous Doppler Nyquist ranges of a few m/s are achieved by GRaWac. Chirp bandwidths of approximately 50 MHz at RF frequencies are used in most observing scenarios. Typically, two to four different chirp sequences are defined in a chirp table prior to operations to optimize sensitivity over all observed ranges. The chirp frequency bandwidth is decreased to yield lower resolution but higher sensitivity further away from the radar. The control of chirp settings, synchronization of chirps, and signal processing is done on an integrated radar PC, which is controlled via flexible software.

The transmitted power is monitored for calibration purposes before combining the two frequencies within the diplexer. Antenna losses are accounted for in post-processing. A low-loss frequency-selective diplexer is used to combine transmission signals and to separate the two frequencies in the receiver.

A low-noise amplifier (LNA) is used before frequency separation to improve sensitivity, which also provides calibration signals for the receiving chains. Relative gain calibration is based on switching the first stage of the receiver LNA during operation (Ogut et al., 2021). For absolute hot/cold calibration on the ground, both liquid nitrogen cold, and ambient hot load references are used, similarly to the well-established W-band radar manufactured by Radiometer Physics GmbH (Mech et al., 2019; Küchler et al., 2017).

A summary of GRaWAC performance parameters and sensitivity is given in Table 1.

#### 135 3 Differential Absorption Radar Methodology

We here revisit the Differential Absorption Radar (DAR) methodology introduced by Lebsock et al. (2015), and previously applied to ground-based (Roy et al., 2018, 2020) and air-borne (Roy et al., 2022; Millán et al., 2024) VIPR measurements. We first describe the profiling retrieval, including a sensitivity analysis to retrieval parameters (Sec 3.1). Then, we describe the IWV retrieval (Sec 3.2), which makes use of the ground return. The retrieval code is publicly available in Schnitt (2025b).

Table 1. GRaWAC technical specifications

| parameter                             | specification   |                 |  |
|---------------------------------------|-----------------|-----------------|--|
| frequency / GHz                       | $167.3 \pm 0.1$ | $174.7 \pm 0.1$ |  |
| wavelength / mm                       | 1.8             | 1.7             |  |
| transmit power / mW                   | 70              | 90              |  |
| gain / dB                             | 54.6            |                 |  |
| receiver noise / dB                   | 5.5             |                 |  |
| receiver intermediate frequency / MHz | 4               |                 |  |
| dynamic range / dB                    | 58              |                 |  |
| antenna diameter / m                  | 0.5             |                 |  |
| beam width / $^{\circ}$               | 0.36            |                 |  |
| power consumption / W                 | 700             |                 |  |
| weight / kg                           | 116             |                 |  |
| dimension / m <sup>3</sup>            | 115 x 90 x 90   |                 |  |

#### 140 3.1 Profiling

A radar's measured reflectivity factor  $Z_e$  at frequency f and range r depends on the scattering properties of the respective radar volume ( $Z_{e,scatt}$ ) and the attenuation of the signal by gas and hydrometeor concentration along the radar's beam path according to Beer's law (Eq. 1).

$$Z_e(r,f) = Z_{e.scatt}(r,f)e^{-2\cdot\tau(0\to r,f)}$$
(1)

The ratio of  $Z_e$  measured at  $r_1$  and a subsequent range gate  $r_2 = r_1 + R$  is then by Eq. 2 for fixed f:

$$\frac{Z_e(r_1, f)}{Z_e(r_2, f)} = \frac{Z_{e, scatt}(r_1, f)}{Z_{e, scatt}(r_2, f)} e^{-2(\tau(0 \to r_1, f) - \tau(0 \to r_2, f))} = \frac{Z_{e, scatt}(r_1, f)}{Z_{e, scatt}(r_2, f)} e^{2\Delta \tau}, \tag{2}$$

with  $\Delta \tau = \tau(0 \to r_2, f) - \tau(0 \to r_1, f) = \tau(r_1 \to r_2, f)$ . The optical depth  $\tau$  is the integral of the extinction coefficient  $k_e$  over distance r:  $\tau(r, f) = \int_0^r k_e dr'$ . Replacing  $\Delta \tau$  in Eq. 2 yields

$$\frac{Z_e(r_1, f)}{Z_e(r_2, f)} = \frac{Z_{e,scatt}(r_1, f)}{Z_{e,scatt}(r_2, f)} e^{2Rk_e(r_2, r_1)}.$$
(3)

The extinction coefficient  $k_e$  is the extinction sum of a gas  $(k_{e,gas})$  as well as a (liquid) hydrometeor  $(k_{e,hydro})$  contribution:  $k_e(r,f) = k_{e,gas}(r,f) + k_{e,hydro}(r,f)$ . The gaseous component  $k_{e,gas}$  comprises a water vapor  $k_{e,v}$  and a dry air  $k_{e,dry}$  component:  $k_{e,gas} = k_{e,v} + k_{e,dry}$ . Extinction  $k_e$  is a product of the density  $\rho$  and the mass extinction coefficient  $\kappa$  of a medium. For water vapor, thus, we can relate  $k_{e,wv}$  with the absolute humidity  $\rho_v$  as  $k_{e,v} = \rho_v \cdot \kappa_v$ . Replacing  $k_e$  in Eq. 3 yields:

$$\frac{Z_e(r_1, f)}{Z_e(r_2, f)} = \frac{Z_{e,scatt}(r_1, f)}{Z_{e,scatt}(r_2, f)} e^{2R \cdot (\rho_v \kappa_v + k_{e,dry} + k_{e,hydro})}.$$
(4)

155  $Z_e(r_1, f)$  and  $Z_e(r_2, f)$  are measured quantities in units of  $mm^6m^{-3}$ . As the eventual goal is to retrieve  $\rho_v$  from measured reflectivities, we invert and re-organize Eq. 4 to obtain the observed absorption coefficient  $\gamma$  in units of  $m^{-1}$ :

$$\gamma(r_1, r_2, f) \equiv \frac{1}{2R} ln\left(\frac{Z_e(r_1, f)}{Z_e(r_2, f)}\right) = \rho_v \kappa_v + k_{e,dry} + k_{e,hydro} - \frac{1}{2R} \cdot ln\left(\frac{Z_{e,scatt}(r_1, f)}{Z_{e,scatt}(r_2, f)}\right), \tag{5}$$

with  $c_{\text{scatt}}(f) = \frac{1}{2R} \cdot ln\left(\frac{Z_{e,scatt}(r_1,f)}{Z_{e,scatt}(r_2,f)}\right)$ . When measuring at two frequencies, a solution for  $\rho_v$  can be found by subtracting Eq. 5  $f_1$  and  $f_2$ :

160 
$$\gamma(r_1, r_2, f_1) - (r_1, r_2, f_2) = \rho_v(\kappa_v(f_1) - \kappa_v(f_2)) + \Delta c_{\text{scatt}} + \Delta k_{e,dry} + \Delta k_{e,hydro},$$
 (6)

with  $\Delta c_{\text{scatt}} = c_{\text{scatt}}(f_1) - c_{\text{scatt}}(f_2)$ , and  $\Delta k_{e,dry}, \Delta k_{e,hydro}$  analogously.

With GRaWAC's two channels located along the wing of the 183 GHz water vapor absorption line, we assume that the frequency dependence of dry air is weak relative to the water vapor contribution, thus  $\Delta k_{e,dry} \approx 0$ . As discussed in Roy et al. (2018) and Millán et al. (2024), the differential scattering term contribution is negligible in homogeneous cloud conditions, but adds a biasing term in the case of large or variable hydrometeor occurrence. The differential hydrometeor extinction term  $\Delta k_{e,hydro}$  is negligible in ice-only clouds, but adds a bias due to liquid attenuation differences if not corrected before retrieval. Neglecting the differential terms leads to a direct expression for deriving  $\rho_v$  from measured quantities at each radar range gate r:

$$\rho_v(r) = \frac{\gamma_r(r, f_1) - \gamma_r(r, f_2)}{\kappa_v(r, f_1) - \kappa_v(r, f_2)} = \frac{\Delta \gamma(r)}{\Delta \kappa(r)}.$$
(7)

When applying the retrieval to GRaWAC measurements, we first calculate  $\gamma_{f_1}$  and  $\gamma_{f_2}$  from the measured Ze at 167.3 and 170 174.7 GHz, using the original radar range resolution fixed by the chirp settings. Starting from each range bin  $r_1$ , we find the bin centered at  $r_2$  which is R away (so  $r_2 = R + r_1$  holds true). As each chirp sequence has a different range resolution (see Sec. 4), the number of radar bins that fit into R varies between chirp sequences and is available for later quality control of the retrieval. Using temperature and pressure information obtained from radiosonde or dropsonde profiles, we determine values of  $\kappa_v$  for each radar measurement by referencing a pre-compiled lookup table. The lookup table was calculated for a large range of temperature and pressure values using the Rosenkranz absorption module (Ori, 2019). While the choice of R manifests the true resolution of the retrieved water vapor profile,  $\rho_v$  is calculated for each measurement time and range bin, respectively. To account for measurement noise, we apply further quality checks and temporal as well as vertical smoothing. First, we flag  $\gamma$ when fewer than three radar bins lie within R, or when the Signal-to-Noise Ratio (SNR) at either channel is below 1 dB. Then, we resample the retrieved water vapor profiles to  $t_{avg}$  by calculating the mean profile within the respective time span. Lastly, we apply a rolling mean filter to smooth the obtained profiles with a window of R. We test the sensitivity of the retrieval to choices of R and  $t_{avq}$  by comparing retrieved  $\rho_v$  to a reference radiosonde profile (Fig. 2) at very low water vapor loading. Note that this case is further discussed below in Fig. 5. We find that the bias to the reference is minimized for R = 200m and  $t_{avg} = 60s$ , and we choose this setting for ground-based deployment, as illustrated in Sec. 4.

When operated from an aircraft, water vapor profiles are retrieved along the axis of the inclined radar  $\theta$ . Both radars are inclined backward with respect to the fuselage to avoid contamination by the strong reflection from the surface. Retrieved

Figure 2. Profiles of (a) reflectivity at 167.3 (orange) and 174.7 GHz (purple), (b) differential absorption  $\gamma$  (black) and extinction  $\kappa$  (blue) coefficients, and difference between GRaWAC  $\rho_v$  and coincident radiosonde profile absolute humidity with (c)  $t_{avg}=60s$  and varying R and (d) varying  $t_{avg}$  and R=200m. Areas of low SNR are flagged in grey. The example shows an ice cloud observed between 3000 and 5000 m at AWIPEV station on February 20, 2025, and is further discussed in Sec. 5.

profiles are projected to nadir according to Eq. 8.

$$\rho_{v,nadir} = \rho_{v,\theta} \cdot \cos(\theta) \tag{8}$$

#### 3.2 IWV

Airborne operation enables the continuous monitoring of total moisture below the aircraft based on the surface reflection. As first introduced by Roy et al. (2022), Partial-column IWV below the aircraft can be retrieved from ground Normalized Radar Cross Section (NRCS)  $\sigma_0$  returns.  $\sigma_0$  is a function of frequency f and incidence angle  $\theta$ , and can be calculated as follows (Roy

et al., 2022):




$$\sigma_0(f,\theta) = Z_{e,\text{ground}} \cdot K_w^2 \cdot \pi^5 \cdot \lambda^{-4} \cdot \cos\theta \cdot \Delta r,\tag{9}$$

with  $K_w = 0.86$ ,  $\lambda$  the wavelength of each GRaWAC channel, and  $\Delta r$  the range resolution at surface. Analogously to Roy et al. (2022), we calculate a reflectivity signal at ground  $Z_{e,ground}$  by finding the maximum  $Z_e$  at ranges between 300 and -300 m distance.  $\sigma_0(f,\theta)$  is calculated at each measurement timestep, and takes the position of the aircraft into account as the pitch angle varies with flight time. Radar inclination angle  $\theta$  is calculated at each measurement time with  $\theta = \theta_{pitch} - \theta_{bellypod}$ , with  $\theta_{bellypod}$  being the backward inclination of the bellypod of 25° with respect to the aircraft fuselage.

Following the argumentation in Roy et al. (2022), IWV can be derived from the observed surface NRCS values according to

$$IWV = \frac{\sigma_{0,f_1}}{\sigma_{0,f_2}} \cdot \frac{\cos \theta}{2 \cdot (\kappa_v(f_2) - \kappa_v(f_1))}.$$
(10)

We calculate  $\kappa_v$  in the same manner as in profile retrieval by forward simulating dropsonde profiles with PAMTRA. According to Roy et al. (2022), if  $\Delta\kappa_v = \kappa_v(f_2) - \kappa_v(f_1)$ ) does not vary significantly between aircraft and ground in observed conditions (see Sec. 5), the actual profile shape cancels out in the IWV calculation. We, thus, assume a constant  $\kappa_v = 0.11 \text{ m}^2\text{kg}^{-1}$ . Using Gaussian error propagation, the expected error due to this assumption is below 0.01 kgm<sup>-2</sup> and lower than expected uncertainties due to measurement noise. Note that, in contrast to the profile retrieval, IWV retrieval is sensitive to the absolute calibration of the radar.

## 4 Ground-based measurements at AWIPEV station, Nv-Alesund

During February and March 2025, GRaWAC was embedded into the remote sensing suite at AWIPEV research base in Ny210 Ålesund, Svalbard, within the Intensive Observation Period for Water (IOP4H2O). The observed period coincided with anomalously warm conditions (Bradley et al., 2025). The radar was installed on the roof of the observatory in the close vicinity of continuous atmospheric measurements, including a HATPRO microwave radiometer, a Vaisala CL51 ceilometer, and a Radiometer Physics GmbH Ka- and W-band cloud radar (e.g., Chellini et al., 2023). GRaWAC measured continuously without major interruptions throughout the observation period. In addition to the triple-frequency radar observations, frequent radiosonde launches were conducted to complement the regular 12-hourly soundings at AWIPEV (Maturilli, 2020).

Throughout the deployment, GRaWAC measured with the chirp settings summarized in Tab. 2. Four different chirp sequences were used similar as in the Ka- and W-Band radar. The vertical resolution was matched to the Ka- and W-band radar resolutions, respectively, as closely as possible, thereby facilitating multi-radar matching. Meanwhile, the Nyquist velocities were maximized to avoid spectral aliasing.

While the observational period will be analyzed statistically elsewhere, we focus here on a selected case study to illustrate GRaWAC's potential for continuous water vapor profiling and for studying microphysical processes. GRaWAC measurements and retrievals of the analyzed case of February 20, 2025, are available in Schnitt (2025a).

Fig. 3 illustrates the conditions on February 20, 2025, between 08:00 and 14:30 UTC. Throughout the case study, clouds were observed at three distinct levels: a high pure ice cloud persisted between 5 and 7.5 km for the first 2 hours and developed

Table 2. Chirp table settings of GRaWac during IOP4H2O in February/March 2025 at AWIPEV station

|                                | GRaWAC    |           |           |            |
|--------------------------------|-----------|-----------|-----------|------------|
| parameter                      | chirp 1   | chirp 2   | chirp 3   | chirp 4    |
| Measuring Range / km           | 0.1 - 0.4 | 0.4 - 1.2 | 1.2 - 3.0 | 3.0 - 11.9 |
| Range Resolution / m           | 16.0      | 17.8      | 23.5      | 28.6       |
| Integration Time / s           | 0.08      | 0.1       | 0.4       | 0.9        |
| Nyquist velocity / $ms^{-1}$   | 4.8       | 4.3       | 2.5       | 1.6        |
| sensitivity at 1km range / dBZ | -43       |           |           |            |

Figure 3. Measurements at AWIPEV station on February 20, 2025. Time-height contour plots of (a) Ze at 167.3 GHz, (b) Ze at 174.7 GHz, (c) SNR at 167.3 GHz (color-coded 1 dB and up) with SNR below 1 marked in grey, (d) DAR 167.3-174.7 GHz, (e) mean Doppler velocity  $v_d$  at 167 GHz, as well as (f) timeline of HATPRO-derived integrated water vapor (IWV; black), liquid water path (LWP; red), and sounding-derived IWV (blue). HATPRO measurements after 13:30 were flagged for precipitation (grey). Gaps in IWV and LWP result from elevation scanning. The time of radiosonde launches is marked by vertical lines (blue shading).

to a higher reflective, thinner layer between 3 and 5 km between 10:00 and 12:30 UTC, while a mixed-phase cloud deck began forming at 11:30, persisting throughout the afternoon. SNR is high throughout the lower cloud layer and the lowest layers of








the upper cloud, but drops below 1 dB at farther distances around 5 km (Fig. 3c). SNR might generally be higher at 5 km in cases of reduced attenuation (due to water vapor and liquid) or larger ice particles.

Moistening is observed in the HATPRO IWV throughout the case, increasing from 3.0 kgm<sup>-2</sup> at 08:00 UTC to 5.0 kgm<sup>-2</sup> at 12:30 UTC. HATPRO and radiosonde-derived IWV match within 0.3 kgm<sup>-2</sup> or better. The mixed-phase cloud contains low liquid water path (LWP) of generally less than 100 gm<sup>-2</sup> (as derived from HATPRO radiometer measurements), precipitates with maximal Ze reaching up to 15 dBZ at 167 GHz, and most likely includes rimed snowfall between 12:10 and 12:30 UTC as seen by the sudden increase in LWP to 300 gm<sup>-2</sup>. Ze at 174.7 GHz is reduced compared to Ze at 167.3 GHz due to attenuation by water vapor and liquid water. The loss of signal in the 174.7 GHz channel due to increased attenuation can be observed, for example, in the thin cloud layer prevailing at 11:30 UTC at 7.5 km. A low SNR of below 1 dB prevails around cloud edges. While the mean Doppler velocity is dominated by negative values between -1 and 0 in the ice-dominated cloud parts, positive updrafts occur frequently in the low cloud layer.

The radiosonde profiles measured at 10:47, 13:02, and 14:02 UTC show that moistening is driven by an increase in water vapor density in the lowest 1 km. Temperature inversions, e.g., at 2 and 4 km, cap the respective mid- and low-level cloud layers. Saturation with respect to ice is reached throughout both cloud layers. The higher temperature inversion is accompanied by a moisture inversion, where saturation with respect to liquid is reached at 13:02 and 14:02 UTC, respectively. Saturation with respect to liquid is also reached throughout the entire mixed-phase cloud deck at 14:02 UTC, hinting at rimed snow conditions at this time, where LWP is increased. At 10:47 and 13:02 UTC, relative humidity decreases from saturation at 900 m to 60-80 % at ground. Cumulative two-way attenuation of GRaWAC's two channels at each of the sounding launches (Fig. 4d)) indicates strong signals and positive gradients in areas of higher moisture contents (below 2.2 km). The increasing total moisture content from 10:47 to 14:02 UTC, at 2 kgm<sup>-2</sup>, corresponds to a 0.3 and 1 dB increase in gas attenuation at 167 and 174 GHz, respectively.

Profiles of reflectivity and DAR sampled 1 minute around each radiosonde launch are depicted in Fig. 5, first and second columns, respectively. The high cloud feature persists at 10:47 and 13:02 UTC, with reflectivities ranging from -20 to -10 dBZ at 167.3 GHz. SNR decreases from more than 5 dB throughout the layer at 10:47 UTC to below 0 dB at 13:02 UTC. The low cloud forming below 2000 m shows higher reflectivity values at 167.3 GHz, ranging from 0 to 10 dBZ, at 13:02 and 14:02 UTC, respectively. The change of DAR signal with height aligns well with the shape of cumulative attenuation due to water vapor shown in Fig. 4(d). High DAR variances (grey areas) are present in areas of low SNR (e.g., Figs. 5(d) above 3 km, 5(f) above 1.2 km) or in areas affected by enhanced differential hydrometeor scattering impact in precipitation-affected vertical levels (e.g., 5(d) at 500 m). At these levels, differential scattering, as well as imperfect antenna overlap, lead to negative DAR values.

We run the water vapor profile retrieval using the DAR approach, as summarized in 3.1, on GRaWAC's measurements. Fig. 5 (c, e, g) show the derived profiles with R = 200 m and 60 s temporal resolution. Note that retrieval outliers, due to SNR below 1 or negative DAR values, are outside the plot range or are flagged in grey. Moisture contents are below 0.7 gm<sup>-3</sup> throughout the upper cloud layer (see radiosonde profile (c)). We thus interpret the fluctuation of the retrieved profile as uncertainty that is expected for the retrieval at its minimal condition. While it seems like the microwave radiometer (MWR)


**Figure 4.** Radiosonde profiles launched at 10:47 (light blue), 13:02 (medium blue), and 14:02 (dark blue) of (a) temperature, (b) absolute humidity, (c) relative humidity with respect to liquid (solid) and ice (dashed), and (d) 2-way bottom-up cumulated gas attenuation at 167.3 (solid) and 174.7 GHz (dashed) obtained with PAMTRA.

profile corresponds well with the RS profile, MWR-retrieved absolute humidity profiles at altitudes above 3 km contain hardly any to no information content, and have a true vertical resolution of less than 500 m (e.g., Walbröl et al., 2022; Schnitt et al., 2024). Strong differential hydrometeor scattering and liquid attenuation effects are present in the derived profile at 13:02 UTC below 1000 m, as indicated by substantially deviating humidity estimates compared to the sounding profile. The further moistened boundary layer (see Fig. 4) at 14:02 UTC is well represented by the DAR retrieved humidity profile. Profiles of DAR and RS agree remarkably well, with deviations of less than 0.5 gm<sup>-3</sup>. Similar deviations were found in Roy et al. (2020).

Although the boundary layer moistening is represented by GRaWAC DAR-derived profiles, further rigorous research is needed to test and thoroughly understand the limitations of the presented retrieval approach. As proposed by Roy et al. (2020) and Battaglia and Kollias (2019) and applied to VIPR measurement by Millán et al. (2024), the addition of a third frequency to the retrieval approach improves the retrieval performance, accounting for effects of differential hydrometeor scattering. As


Figure 5. Evaluation of three retrieved water vapor profiles at 10:47 UTC (first row), 13:02 UTC (second row), 14:02 UTC (third row) with profiles of (a, d, g) mean reflectivity (solid) and SNR (dashed) at 167 (purple) and 175 (orange) GHz, (b, e, h) mean DAR 167-175 (black), and (c, f, i) retrieved absolute humidity profiles from GRaWAC (black) with R = 200m and  $t_{avg} = 60s$ , and microwave radiometer (MWR) (cyan) as well as sounding absolute humidity profile (blue). Grey dots in (c), (f), and (i) are quality flagged. Shaded areas in (b,e,h) correspond to the standard deviation of the respective measurement within the averaging time of the water vapor retrieval. Non-complete overlap is marked in grey. Note the different height ranges in both rows centered around the respective cloud appearance.

GRaWAC is limited by its design to two frequencies, incorporating a third frequency into the present retrieval approach is not possible. Simultaneous Ka- or W-band measurements may be too distant from deployed frequencies (personal communication, M. Lebsock), as scattering processes differ due to the different wavelength regimes.

We therefore hypothesize that the current retrieval is not the best fit for dry conditions, given the complex mixed-phase microphysics that is ubiquitously observed in the Arctic. An optimal estimation approach that includes passive MWR as well as W-band cloud radar measurements would be better suited to account for differential scattering, improve the retrieval's information content, as previously demonstrated by Schnitt et al. (2020) for the tropical atmosphere, and further enhance understanding of retrieval uncertainties. Pairing the water vapor retrieval with a Particle Size Distribution (PSD) retrieval would further enable the simultaneous quantification of cloud microphysics and water vapor in a synergistic framework. Rather than only examining the moments of the Doppler spectrum, the actual spectra can provide more information on the microphysical



Figure 6. Like in Figure 4, but between 11:45 and 12:30.

situation within the cloud (e.g., Shupe et al., 2004). We illustrate here GRaWAC's spectra measured at 167.3 GHz by zooming in on a transition case between 11:59 and 12:20 UTC (Fig. 6). In the transition, MWR recorded LWP increases from 17.5 to 46.0 and 277.4 gm<sup>-2</sup> at 11:59, 12:04, and 12:20 UTC, respectively. While the upper cloud layer thins slightly throughout the period, the lower cloud layer develops strong Ze signals of up to 15 dBZ. Precipitation sublimates in the sub-cloud layer below 600 m, as reflectivities decrease towards the ground.

Fig. 7 shows the Doppler spectra recorded at 3500, 1250, and 990 m at the three times, respectively. The upper cloud layer is dominated by downward motions between -1.5 and -3.0 ms<sup>-1</sup>, which correspond to ice hydrometeors and increase in fall speed with time. In the upper part of the lower cloud layer, at 1250 m (panel (b)), the dominating spectral peak transitions from liquid-only at 11:59, ice-only at 12:04, to mixed-phase at 12:20 UTC. The same feature can be observed further down the cloud layer at 990 m (panel (c)), where, however, liquid and ice peaks show a more distinct separation and a narrower spectral width.

Future work will encompass further micropyhsical studies making use of the triple-frequency radar setup operational during IOP4H2O. Dual-frequency ratios, as well as Differential Doppler Velocities, will be used to retrieve particle sizes and detect pockets of supercooled liquid (Korolev, 2007). While no further Mie notches were present in the presented case, we speculate that vertical air motions could be rather retrieved from GRaWAC spectra in liquid precipitation, using the approach by Courtier

**Figure 7.** Doppler spectra at 167 GHz averaged to 10 s resolution at (a) 3500 m, (b) 1250 m, (c) 990 m, recorded at 11:59 (purple), 12:04 (cyan), and 12:20 UTC (magenta).

et al. (2024). Future work will also investigate the feasibility of the IWC retrieval presented in McCusker et al. (2025) for frequencies at 167 GHz.

# 5 Airborne deployment

GRaWAC was installed aboard the Alfred Wegener Institute's Polar 6 (Wesche et al., 2016) aircraft during the Humidity profiles and Arctic Mixed-phase clouds as seen by Airborne W- and G-band radars (HAMAG) campaign. As shown in Fig. 8(a), the aircraft was equipped with a sub-millimeter MWR (Mech et al., 2019), operated from within the cabin, and a bellypod carrying GRaWAC and the W-band radar Microwave Radar/radiometer for Arctic Clouds (MiRAC; Mech et al., 2019). Both radars were installed with the same backward inclination angle of 25° with respect to the aircraft fuselage for a matched field of view.

Dropsondes were frequently launched to sample profiles of temperature, humidity, pressure, and wind. Operating from Kiruna,


2000

1750

1500

1250 E

Elevation 5

500

250

100

80

20

**Figure 8.** (a) AWI Polar 6 aircraft equipped with dropsondes, microwave radiometer, and dual-frequency W-G radar configuration in bellypod, (b) map of HAMAG research flights (color) with mean sea ice conditions February 2025 (blue contours) and orography (grey contours).

Sweden, six research flights (RF) took place between February 7, 2024, and February 22, 2024 (see Fig. 8(b)). Local flights around Kiruna were performed to test the instrument configuration (RF02-RF04). These flights were dominated by clear-sky scenes with low-level fog visible in both radars reaching up to 200 m. Cellular convection originating from a cold air outbreak was sampled along the Norwegian coast off Tromsø (RF05). The evolution of clouds and precipitation along a sea-ice to open-ocean transect was investigated during RF01 and RF06. Due to software issues, DAR could not be analyzed during RF01. All measurements are available in Mech et al. (2024).

Different chirp settings were tested in RF01-RF03 to run sensitivity and software tests. Selected chirp settings (summarized in Tab. 3) were different than those during ground-based deployment to account for the effects of the moving platform. The maximum range of the first chirp was chosen to match the zone of imperfect overlap. As MiRAC's settings had been successfully operational in previous deployments (e.g., Mech et al., 2022; Ehrlich et al., 2025), GRaWAC's range resolution was attempted to match as best as possible to facilitate dual-frequency matching.

Several post-processing routines are necessary to optimize dual-frequency matching to obtain optimally matched dual-frequency ratios (DFR). Range matching is achieved by calculating the means of MiRAC Ze within each GRaWAC bin. By calculating exact time stamps at the end of each chirp (rather than at the end of the overall chirp sequence as stored in the

Table 3. Chirp settings during HAMAG RF05 and RF06

|                                    | GRaWAC    |           | MiRAC-A   |           |
|------------------------------------|-----------|-----------|-----------|-----------|
| parameter                          | chirp 1   | chirp 2   | chirp 1   | chirp 2   |
| Measuring Range / km               | 0.1 - 0.6 | 0.6 - 4.0 | 0.1 - 0.6 | 0.6 - 4.0 |
| Range Resolution / m               | 16.0      | 26.7      | 4.5       | 13.5      |
| Integration Time / s               | 0.6       | 0.5       | 0.4       | 0.4       |
| Nyquist velocity / ${\rm ms}^{-1}$ | 6.7       | 5.8       | 16.3      | 6.9       |
| sensitivity at 1 km range / dBZ    | -43       |           | -43       |           |

instrument output data), MiRAC and GRaWAC volumes are matched, similar to previously applied in Chellini et al. (2023). In addition to range and time matching, DFR signals were corrected for gas attenuation to eliminate atmospheric-related signals from the differential radar signals. We use the Passive and Active Microwave TRAnsfer simulator (PAMTRA; Mech et al., 2020) to obtain gas attenuation profiles based on dropsonde profiles. Dropsonde profiles were complemented by ERA5 (Hersbach et al., 2020) reanalysis profiles above the aircraft. Cumulated two-way attenuation profiles at dropsonde launch times were temporally interpolated to match the time resolution of the radars and were added to the measured DFR for correction.

In the following subsections, we first illustrate a case of a retrieved water vapor profile and evaluate retrieved IWV based on the methods introduced in Sec. 3.2. Then, we investigate the added value for observing cloud microphysical processes based on a seeder-feeder cloud case observed with the dual-frequency configuration during RF06.

# 5.1 Water Vapor Profile and amount

We illustrate GRaWAC's airborne water vapor profiling capabilities based on measurements obtained on February 18, 2024, during RF05 off the coast of Tromsø, Norway. During RF05, Polar 6 sampled a square along the coast twice at a flight altitude of around 3000 m.

Fig. 10 illustrates GRaWAC's Ze and DAR measurements as well as MiRAC's W-band and W-G dual-frequency observations around a dropsonde launched at 10:32:30 UTC. Note that Ze, DFR, and DAR profiles here are not at nadir, but correspond to observations along the slanted path at the inclination angle  $\theta$ . Attenuation effects due to water vapor and liquid attenuation reduce G-band reflectivity compared to W-band Ze at the time of launch. DAR signals reach 4.8 dB at ground. The retrieved IWV agrees well with the IWV derived from the dropsondes.

We retrieve water vapor profiles at 20 s temporal resolution, and R = 200m using the profile retrieval outlined in Sec. 3.1. Water vapor profiles have been corrected to nadir using Eq. 8. With ground speeds of around 80 ms<sup>-1</sup>, the horizontal resolution of the water vapor profile is around 1.5 km. GRaWAC and dropsonde profiles match remarkably well within 0.5 gm<sup>-3</sup>. Improvements compared to the ground-based performance are most likely due to reduced attenuation effects at low ranges close to the aircraft (compared to at ground, where attenuation is maximal close to GRaWAC), and lower variation of Ze and DAR input signals due to more homogeneous conditions.

350

355

**Figure 9.** Time-range contour of (a) MiRAC and (b) GRaWAC Ze, (c) attenuation-corrected W-G DFR, (d) GRaWAC DAR and (e) dropsonde (blue) and GRaWAC (black) IWV during RF05, 18.02.24. Time of dropsonde launch is marked in blue.

In order to evaluate the IWV retrieval on a statistical basis, we compare GRaWAC IWV to dropsonde IWV during all flights (Fig. 11). GRaWAC IWV represents the mean IWV within a one-minute window around each respective dropsonde launch.

All seventeen dropsondes were dropped in cloudy conditions. Low standard deviations within the 1-minute averaging window signal that conditions were mostly homogeneous within the averaging time. Retrieved IWV values are highly correlated to dropsonde IWV (correlation coefficient 0.79), and show a root-mean square difference of 1.1 kgm<sup>-2</sup> with a bias of 0.3 kgm<sup>-2</sup> in comparison. The RMSD observed here is slightly higher than the value found in Roy et al. (2022), who find an RMSD of clear-sky IWV compared to dropsonde IWV of 0.8 kgm<sup>-2</sup>. RMSD is lower than in Millán et al. (2024) who find a RMSD of 1.5 kgm<sup>-2</sup> in predominantly cloudy conditions after modifying VIPR's calibration. Occurring biases are caused in part by differential hydrometeor extinction, which according to Roy et al. (2022) can even lead to differences of up to 5 kgm<sup>-2</sup>. Another source of error is the calibration as the IWV retrieval is sensitive to the absolute calibration of the radar. As the IWV derived during RF04 is substantially stronger and more biased than during all other flights, we believe that these offsets are driven by calibration-induced uncertainty. When excluding RF04 sondes, the bias reduces to -0.3 kgm<sup>-2</sup>. RMSD between GRaWAC and dropsonde IWV is higher over the ocean (0.4 kgm<sup>-2</sup>, N=5) than over land (0.3 kgm<sup>-2</sup>, N=7) or ice (0.4 kgm<sup>-2</sup>, N=5),

Figure 10. Profiles recorded at 10:32 UTC during RF05 of (a) Ze at 167.3 (purple) and 174.7 GHz (orange), (b) DAR, (c) dropsonde-derived (blue) and GRaWAC (black) retrieved  $\rho_v$  with R=200 m. Profiles in all panels represent mean conditions averaged to 20 s. Standard deviation within  $t_{avg}$  is given by the shaded area.

respectively. We attribute the differences in RMSD between land/ice and ocean in part to a sampling bias, given that flights over land were often dominated by low-level fog conditions that reached altitudes of 200 m or less, while clouds over the ocean reached cloud top heights of 2.5 km, with occurrences of differential scattering and liquid attenuation. Our results suggest that space-borne DAR-derived IWV over snow- or ice-covered surfaces could outperform passive MWR retrievals suffering from uncertainties related to unknown surface emissivities. Yet, more measurements over land and ice surfaces with thicker clouds are needed to critically assess the true benefits.

### 5.2 W-G dual-frequency: seeder-feeder cloud case

RF06 sampled cloud evolution over Bothnian Bay on February 19, 2024, along a transect reaching from sea ice at the Northern edge down to 61.57°N and 19.21°E over open ocean (see Fig. 8(b)). Dropsondes sampled the thermodynamic evolution along the transect. No clouds were observed along the transect over sea ice until the first cloud signals started forming at the sea ice edge.

Figure 11. Retrieved GRaWAC versus dropsonde-derived IWV over ice (dot), land (square), and ocean (triangle), color-coded by research flight.

Fig. 12 illustrates the cloud deck sampled between 11:20 and 11:45 UTC around the turning point, with a dropsonde launched at 11:24 UTC. Note that the same cloud was sampled twice, before and after turning, respectively. From ice to open ocean, the boundary moistens, as indicated by the increasing IWV from 7.0 kgm<sup>-2</sup> to 9.8 kgm<sup>-2</sup> at the turning point. In phases of constant aircraft motion (before 11:24), IWV standard deviation within one minute is 0.3 kgm<sup>-2</sup>, which we interpret as the precision of the retrieval.

While Ze measurements at W- and G-band both capture seeding fallouts from the upper cloud layer, a clearer picture of the present micro-physical processes develops when analyzing the DFR signal (panel (c)). Low DFR signals around zero dominate the feeding upper cloud, indicating small ice hydrometeors with sizes that fall within the Rayleigh scattering regime in both W- and G-bands, respectively. DFR signals around 0 at cloud top also hint at good inter-calibration of the two radars. Interestingly, DFR signals at 1000 m range do not increase even when Ze in both radars increases in a thickened upper cloud.

Low DFR signals are also present in the upper part of the seeded cloud. Increasing hydrometeor sizes, represented by increasing DFR signals, only appear beyond 2000 m in range corresponding to a vertical level of approximately. 900 m above ground. This vertical level is remarkably distinct and stable along the sampled cloud field extent. Temperature conditions at this level are around -5°C. DFR signals then increase towards the ground as expected, as precipitation forms.

**Figure 12.** Time-height contours of attenuation-corrected (a) W-band Ze, (b) G-band Ze at 167.3 GHz, (c) W-G DFR, (d) GRaWAC IWV, (e) time-dependent aircraft roll (black) and pitch (blue) angles as well as altitude (red). Note that the range of 0 m corresponds to Polar 6 altitude in panels (a-c).

#### 6 Conclusions



We introduce the novel G-band Radar for Water vapor and Arctic Clouds (GRaWAC) instrument by illustrating first ground-based and air-borne measurements from the Arctic. GRaWAC is a worldwide, unique, DAR-capable, Dopplerized FMCW stand-alone cloud radar that simultaneously transmits and receives at 167.3 and 174.7 GHz. Measurements are performed at a vertical resolution of 20 m or less and a temporal resolution of 2 seconds, resulting in a sensitivity of -43 dBZ at a 1 km range. Based on GRaWAC's two channels, the DAR methodology is applied to derive in-cloud water vapor profiles from both ground and aircraft, as well as partial column IWV between aircraft and ground. We use the two-frequency retrieval introduced by Lebsock et al. (2015) and applied in Roy et al. (2018) and Millán et al. (2024). We test the retrieval sensitivity to vertical

by Lebsock et al. (2015) and applied in Roy et al. (2018) and Millán et al. (2024). We test the retrieval sensitivity to vertical resolution (R) and temporal averaging  $(t_{avq})$  settings by comparing the retrieved profile to a reference radiosonde. We find






that setting R = 200 m and  $t_{avg} = 60$  s both enable the vertical and temporal resolution needed to investigate water vapor variability at a satisfying resolution, and reduce the uncertainty due to measurement noise and signal fluctuation. Compared to VIPR, similar RMSD is achieved with a higher temporal resolution. Note that the analysis of cloud microphysics is performed on GRaWAC's original temporal resolution of 2 s. We illustrate GRaWAC's potential and limitations for Arctic clouds and water vapor by analyzing first ground-based and air-borne G-band measurements. Measurements were performed at AWIPEV station, Ny-Ålesund, within the "Intensive Observation Period for Water" (IOP4H2O) in February 2025, and aboard AWI's Polar 6 aircraft within the HAMAG campaign in Northern Scandinavia in February 2024.

We analyze a ground-based case from February 20, 2025, to highlight the potential of water vapor profiling compared to MWR-derived soundings. While we find that the moistening of the lower boundary layer within the investigated period is well represented in the DAR profiles, deviations compared to simultaneous radiosonde profiles of up to 2 gm<sup>-3</sup> are omnipresent due to differential liquid attenuation and scattering effects. Rigorous quality control is necessary to eliminate and mitigate these effects resulting from complex microphysical processes in the retrieval. As modifying the present retrieval approach to three (Millán et al., 2024) or more (Battaglia and Kollias, 2019) channels is technically not feasible due to the system design, we argue that a synergistic optimal estimation-based retrieval would be more suitable for complex mixed-phase cloud conditions in the dry Arctic climate by retrieving water vapor and microphysical parameters in a joint retrieval. Simultaneous observations of high-resolution water vapor profiles and cloud micro-physical properties will advance process understanding in mixed-phase cloud evolution such as obtained from the synergistic remote sensing suite deployed during the MOSAiC (Shupe et al., 2022) drift (e.g., Jimenez et al., 2025; Seidel et al., 2025). The first-ever recorded Doppler spectra at 167.3 GHz track the temporal evolution of a mixed-phase cloud deck at high resolution, indicating the expected bi-modal peak related to the simultaneous presence of supercooled liquid and ice. Existing retrievals of micro-physical quantities which include the multi-frequency setup of Ka-, W- and G-band measurements (e.g., Yurk et al., 2025; McCusker et al., 2025), will be evaluated in a follow-up publication. The analysis of measurements collected during two ship-borne deployments aboard the RV Polarstern as part of the PS 144 (ArcWatch-2) and PS 149 (ArcWatch-3) will shed further light on the microphysical and thermodynamical environment of Arctic mixed-phase clouds.

Thanks to its versatile design, GRaWAC can easily be installed aboard AWI's Polar 5/6 aircraft. A bellypod carries the dual-frequency W-G configuration consisting of MiRAC (Mech et al., 2019) and GRaWAC. During the HAMAG campaign, six research flights with varying radar settings were performed out of Kiruna, Sweden, covering the local area, the Bothnian Bay, and the Atlantic coast off Tromsø. Microwave radiometer measurements, as well as in total 17 launched dropsondes complemented the dual-radar configuration. Air-borne water vapor profiles and IWV were derived using the DAR technique introduced in Roy et al. (2022). To account for the moving platform and, thus, more variable cloud and water vapor scenes, we adapted the retrieval setting to R = 200 m and  $t_{avg} = 20$  s. DAR-derived profiles agreed well with a simultaneous dropsonde profile with deviations of less than 0.5 gm<sup>-3</sup>. A statistical analysis of GRaWAC and dropsonde-derived IWV (N = 17) revealed an RMSD of 1.1 kgm<sup>-2</sup> between GRaWAC and dropsonde IWV averaged over all surface types. Excellent agreement is achieved between GRaWAC and dropsonde IWV (RMSD of 0.4 kg m<sup>-2</sup>, N = 5) over ice. These results suggest that DAR-derived IWV could improve space-borne IWV estimates obtained from passive microwave radiometry, where retrievals are

https://doi.org/10.5194/egusphere-2025-5563 Preprint. Discussion started: 19 November 2025

© Author(s) 2025. CC BY 4.0 License.




largely challenged by unknown surface emissivities. The potential of air-borne W-G dual-frequency measurements for microphysical processes is highlighted by assessing dual-frequency ratios (DFR) measured in an example of a seeder-feeder cloud case. Low DFR signals of  $\approx 0$  in the seeding and upper feeding cloud parts reveal the presence of small hydrometeors in large portions of the cloud. Increasing DFR signals below a remarkably sharp and spatially homogeneous transition level signal the formation of precipitation and hydrometeor growth. Further research flights, planned for spring 2026 in Fram Strait as part of the COMPEX study, will expand the statistics of IWV over ice and the challenging marginal sea ice zone, and will investigate properties of low-level stratus clouds that ubiquitously cover the Arctic sea ice.

Code and data availability. All processing and analysis software is available in Schnitt (2025b). GRaWAC measurements and water vapor retrievals for February, 20, 2025 at AWIPEV station are available in Schnitt (2025a). HAMAG airborne data is available in Mech et al. (2024).

Author contributions. SaS is the main author of the manuscript, developed and performed the retrieval, and prepared all figures. She is the scientific lead of GRaWAC. SaS, SC, and MM conceptualized the manuscript. SC was the lead author of the proposal that led to the funding of GRaWAC. MM was responsible for the HAMAG campaign, and, together with SaS, designed the research flights. JG and TR designed and conceptualized and built GRaWAC at Radiometer Physics GmbH. All authors contributed to the manuscript.

440 Competing interests. The contact author has declared that none of the authors has any competing interests.

Acknowledgements. GRaWAC was funded as a Deutsche Forschungsgemeinschaft (DFG, German Research Foundation) large infrastructure instrument – Projetknummer 502048393. SaS research, as well as logistics costs for IOP4H2O, were funded within subproject E03 in the framework of TRR 172, the Transregional Collaborative Research Center "ArctiC Amplification: Climate Relevant Atmospheric and SurfaCe Processes, and Feedback Mechanisms (AC)3" funded by the DFG – Projektnummer 268020496. SaS would like to thank AWI Potsdam for fruitful discussions during her guest scientist stay. The authors would like to thank the HAMAG Polar 6 crew, Alan Gilbertson, Shannon Sorg, Dwayne Bailey, Martin Gehrmann, Cristina Sans i Coll, Maximilian Stöhr, Pavel Krobot, Linnu Bühler, Nils Risse, Christian Buhren, Friedhelm Jansen, and Uwe Raffalski for their technical support and measurements. The aircraft campaign HAMAG, including flight hours and aircraft operation, has been funded by the Alfred Wegener Institute Helmholtz Centre for Polar and Marine research. We thank the AWIPEV station team for support in the operation and installation of GRaWAC during IOP4H2O. We would like to particularly thank Marino Maturilli and Christoph Ritter at AWI for their support in our measurements, as well as Christian Buhren, Andreas Walbröl, and

Linnu Bühler for their on-site operation of GRaWAC during IOP4H2O. We acknowledge AWI's support, which made our AWIPEV stay within project AWIPEV\_0043 (RiS number 12554) possible.

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
