# Peer review of "G-band Radar for Water vapor and Arctic Clouds (GRaWAC): novel insights on Arctic water vapor, clouds and precipitation"

_EGUsphere, 2025_

## Referee Comment (RC1)

**Review of GRaWAC paper for AMT**

January 16, 2026

**Overall impression**

This paper describes a new radar system "GRaWAC". This a G-band radar, of which there are only 3 or 4 worldwide, so that in itself makes the instrument relatively novel. The system is developed for Arctic field observations, and can be operated on the ground or on an aircraft, with examples of both shown here. Observations at two closely spaced frequencies near the 183 GHz water vapour absorption line allow retrieval of water vapour profiles in a similar manner to the technique previously demonstrated by Roy, Cooper, et al with the VIPR system at JPL, but this time in Arctic environments, which bring their own challenges (i.e. not much vapour!) and examples of such high-latitude retrievals are shown. The paper is in scope for AMT since it describes a new instrument which is of significant interest to the community and illustrates the kind of measurements it is capable of making. So I would like to see this paper published. However, I do think there are problems with the manuscript in it's current form which make it less impactful and convincing than it could be. So my suggestion would be major revision, to give the authors time to reflect on these issues and refine what is presented (assuming they agree with the comments below)

**Primary comments**

The paper does 3 things:

1. It describes the radar itself, with a moderate amount of technical detail

2. It describes the methodology for water vapour retrieval

3. It applies that methodology to illustrative cases, one measured from the ground at Svalbard, and others measured from an aircraft.

Item 2 has been established in previous papers elsewhere, and the authors are largely reviewing/reworking this technique here. Items 1 and 3 are new information to the literature, and are therefore the really important bits, and I would like to see a greater depth to the discussion of both of these elements, but in particular to the analysis of the observations.

**For item 1 (the radar instrument part):**

- Currently there is no description of the data acquisition and processing. How is reflectivity computed? Are the samples used to compute Doppler spectra, from which reflectivity is derived? Or incoherent averaging of received power? Is the background noise level subtracted? Any further filtering? These details are relevant for the water vapour retrieval application because to do profiling the dual frequency ratio (and therefore each reflectivity) needs to be measured to very high precision.

- Are other parameters measured / stored? I see Doppler velocity later on. What about spectrum width, or Doppler spectra, or I&Q samples? Useful to know the full capabilities.

- The radar is FMCW which can result in "range sidelobes". Have you any information about the magnitude of these artefacts in GRaWAC, or any analysis of what kind of situations it is significant in? It looks like perhaps there is a hint of this in Figure 9b (horizontal line of low reflectivity at around 1200m range) and Figure 12a (thin line of low reflectivity at around 3000m, mirroring the echo from the surface)

- You note that there is an issue with the beams not overlapping at short range, and that this loss is calculated with some assumptions. Can you explain in more detail how this is corrected and the likely uncertainties? Also, do you worry at all about near-field effects?

- Some practical information about operation of such an instrument in Arctic environments, and from aircraft would be useful, even if it is brief. I appreciate some of this may be similar to the approaches used previously for the W-band RPG instrument which is very successful, but worth mentioning, and noting anything that needed to be changed or tuned for operation at these higher frequencies.

- You say in a couple of places that an extra frequency would help disentangle other differential effects (scattering etc), but that dual-frequency is

**For item 3 (the case studies)**

- I think the paper would be much more impactful if the water vapour profiling was analysed and interpreted in greater depth. At the moment, we have estimates of $\rho_v$ based on the difference between dual frequency ratio samples spaced 200m apart. These are then compared to radiosonde profiles. While the overall magnitudes of $\rho_v$ are broadly comparable, the structure of the vapour profile is not particularly well correlated between the two measurements, and this did not give me great confidence in the results. I therefore found it confusing to read in several places that the profiles agree "remarkably well". So there are a few things to consider here:

  - First, the authors need to put error bars on their $\rho_v$ profiles. This should not be difficult. You can predict the precision of a reflectivity measurement from the number of chirps, the spectrum width, and the signal to noise ratio (some care required in how this latter quantity is defined). If spectrum width is not available, a "representative" value could be chosen. The point here is that at the moment you can't assess whether the sonde and radar profiles are comparable, because you don't have uncertainties on either of them. So the extent to which agreement is good or bad, and how surprising or remarkable that is, can't be assessed. Once this is done you can go back through and have some more meaningful discussion / evaluation of the comparison and structures obtained.

  - Related to this, there is currently a lot of emphasis in the paper about differential scattering and presence of liquid attenuation influencing the results. And in the discussion you say that these effects lead to "omnipresent" deviations (which you imply to be errors) of up to 2 grams per cubic metre, which seems quite large - this is pretty much the full magnitude of the signal you're trying to retrieve in some cases. It's not clear to me that this differential scattering & liquid attenuation is really a serious effect? How big is differential scattering likely to be in different scenarios - can you show some calculations with PAMTRA? 167 vs 175 GHz is less than 5% difference in frequency, so I would (perhaps naively) expect the difference in reflectivity samples to be quite modest. And for liquid attenuation - again you can evaluate this - but I find it hard to imagine that there is a big *differential* effect here, and of course is only restricted to narrow regions of the cloud where significant LWC exists. But without quantifying these things, I don't think you can interpret differences between retrieval and sonde as being caused by those effects.

  - Related to the vertical structure the $\rho_v$ retrieval: the precision of the profile depends on the precision of the reflectivity estimates, which in turn gets better as you average more chirps. Have you done this experiment? Particularly with the ground based measurements - why not average more, and get a more precise profile which you can have high confidence in? Then you can really compare in detail with the sonde.

  - You could also discuss more clearly about the impact of different vertical resolutions for the retrieval. I didn't feel that choosing this spacing because it minimised the "bias" between radar retrieved profile and radiosonde profile was necessarily a good reason. There is a trade-off here - coarser resolution has a bigger difference in DFR between the two samples, which is easier to measure = more precise estimate. But coarser resolution means loss of detail where there are sharp vertical features to resolve (e.g. airmass boundaries, inversions). It would be good to discuss this aspect more fully, and show what the retrievals look like for different choices, rather than just a residual (taking into account also the next comment - i.e. the radiosonde is not the truth!)

  - I think it's important to acknowledge when comparing sonde to radar that the radar is a true vertical profile, integrated over some well-defined time/length scale; the radiosonde profile is collected over tens of minutes, during which time the sonde drifts by 10s or even 100s of kilometres (you can probably

estimate what this scale is for your cases from the wind profile). So in that sense the radar is "better", or at least better defined in what it represents; and also it means that any comparison between the two is confounded by the fact that they are sampling different parts of the atmosphere - this needs some discussion.

**Secondary comments**

1. Line 96-99 I think there is some repetition here about the bistatic antennas, please have another look

2. Your block diagram mentions frequency multiplication and amplification. I guess the multiplication is ×24? Can you include that somewhere? Optionally, you could break down the multiplication and amplification stages.

3. There is some discussion about calibration of the radar receiver chain by passively viewing black-body targets. What about the transmitter chain?

4. In Table 1, units are indicated in the form "wavelength / mm". Personally I find this notation ambiguous, and would prefer to see the units in brackets, or a separate column of the table. Also note that the units for the dimension are cubic metres but the dimensions provided are centimetres. Also it would be good to replace "gain" with "antenna gain"

5. Line 47 I would move the definition of optical depth slightly earlier, because you need it to understand equation (1)

6. Line 165 - what is "variable hydrometeor occurrence" and why is it important in this context?

7. Line 200 "argumentation" - change to "argument" (I think!)

8. Does equation 10 rely on the ground echo being independent of frequency?

9. Figure 3: can you add more axis labels on the height axis, and consider a background grid to help the reader locate the features you discuss in the text. You might also consider some annotation to help make it easier (e.g. to find the "mixed-phase cloud deck" on line 225 {what altitude?}, or the "thin cloud layer" on line 235)

10. Also figure 3, can you refine the colour scales to bring out more detail. SNR range is 0-40dB but I can't see any features above ~20, and it's hard to see what's going on. Reflectivities seem to be all >-30, so no need to go all the way down to -50 on the current scale. For DAR (a key measurement for your paper's focus) the only features in the yellow colour are ones which are effected by very low SNR, which I would suggest you filter out from this panel, and then tighten the scale down to 0-10dB or something like that (based on the 1D profiles in Figure 5)

11. Line 240 should "Saturation" be "Supersaturation"?

12. Line 268 - you talk about Boundary Layer moistening - where is the Boundary Layer in this case (and how diagnosed?)

13. Figure 4 - useful to add more minor ticks/axis labels on x-axes of these panels and perhaps grid lines

14. Line 290 and thereabouts: you interpret these spectra in terms of different peaks - how did you diagnose what hydrometeors are dominating each peak?

15. Figure 12 can you change the range axis to a height axis, based on the altitude of the aircraft? I found this figure difficult to interpret.